# NEURALIGNER: SCALABLE AND ROBUST DNA SEQUENCE ALIGNMENT VIA EMBEDDING-BASED SIMILARITY SEARCH

## ABSTRACT

DNA sequence alignment is a fundamental task in genomics. Existing aligners rely on the seed–chain–align paradigm, which achieves high efficiency but struggles with sequencing errors and genetic variation. Moreover, most methods remain CPU-based and are poorly suited to large-scale GPU acceleration, limiting their utility in time-sensitive settings. In this paper, we present NeurALigner (NAL), a GPU-accelerated alignment framework that integrates DNA sequence models with vector database retrieval. Instead of exact string matching, NAL encodes DNA subsequences into embeddings and reformulates seed matching as a fast similarity search in feature space, providing robustness to mismatches caused by sequencing errors or genetic variations. The learned embeddings enable the use of longer seeds, raising specificity in matching and improving efficiency. Furthermore, an adaptive seeding strategy dynamically adjusts the number of seeds, balancing efficiency and accuracy. Together, these innovations enable scalable, mismatch-tolerant alignment with high specificity and strong GPU performance.

## 1 INTRODUCTION

DNA is the fundamental blueprint for life, holding the genetic instructions essential for most living organisms. The genetic information is deciphered using high-throughput DNA sequencing technologies (Shendure & Ji, 2008; Rhoads & Au, 2015; Wang et al., 2021), which generate vast numbers of short DNA fragments, referred to as reads. These reads are significantly shorter than the complete genome, which can span from millions of base pairs in bacteria to billions in humans. Therefore, accurate and efficient alignment of short reads to a known reference genome, a process known as read alignment (Li & Homer, 2010), is an essential prerequisite that underpins the interpretation of genetic information and a wide range of downstream applications. For example, in clinical genomics, accurate alignment allows for the detection of disease-related mutations, which supports early diagnosis and informed therapeutic decisions (Yang et al., 2013; Network, 2008). Furthermore, in microbiology, read alignment facilitates the characterization of pathogen genomes, which is crucial for tracking infectious outbreaks and informing effective public health responses (Quick et al., 2016).

Early alignment methods such as Needleman-Wunsch (Needleman & Wunsch, 1970) and Smith-Waterman (Smith et al., 1981) provided exact alignment results but quickly became computationally infeasible as genomic data grew in scale. To address this, heuristic approaches like BLAST (Altschul et al., 1990) introduced efficient search strategies that enabled one-versus-many sequence queries across large databases. However, BLAST is not efficient enough to support the many-to-one alignment required for high-throughput sequencing. To meet these demands, tools such as BWA (Li & Durbin, 2009) and Minimap2 (Li, 2018) were developed and optimized for mapping large collections of reads to a large reference genome. These aligners generally follow a *seed–chain–align* workflow: the reference genome is first segmented and indexed for rapid querying; candidate regions are identified by locating exact-match seeds and chaining them into candidate alignment regions, followed by base-level alignment permitting mismatches and gaps.

Despite these advances, three challenges remain. First, sequencing errors and genetic variations (Logsdon et al., 2025) introduce discrepancies between query reads and the reference genome. Seed-and-extend methods such as BWA and Minimap2 often struggle in these cases because their seeding step requires exact matches, which breaks down when mismatches are frequent. Second, repetitive genomic regions (which make up more than 50% of the human genome) cause the same query fragment to map to multiple locations, leading to alignment ambiguities and errors. Third, although some aligners have been adapted for GPU acceleration (Pham et al., 2023; Sadasivan et al., 2023), most remain CPU-based since the string matching paradigm is not well suited to GPU parallelism. As a result, achieving both high accuracy and efficient GPU-accelerated alignment remains an open problem, yet it is critical for time-sensitive diagnostic workflows in clinical practice (Charalampous et al., 2019; Gu et al., 2021; Brändl et al., 2025).

To address these challenges, we propose **NeurALigner** (**NAL**), a GPU-accelerated sequence alignment framework that integrates DNA models with vector database retrieval (Johnson et al., 2019; Malkov & Yashunin, 2020). Instead of relying on exact string matches, our method encodes DNA sequences into numerical embeddings and reformulates seed matching as a fast, mismatch-tolerant similarity search in embedding space. Trained on large collections of DNA fragments randomly sampled from the reference genome, the model preserves sequence similarity in the feature space and is robust to sequencing errors and genetic variation. The embedding-based similarity search is further accelerated by vector databases. The strength of the DNA model allows us to use longer and fewer seeds, which improves alignment specificity while lowering computational cost. In addition, we introduce an adaptive seeding mechanism that adjusts the number of seeds according to sequence complexity: highly unique sequences require only a few seeds, whereas low-specificity regions trigger additional seeds to maintain accuracy.

In summary, our NeurALigner is a fast and robust DNA sequence alignment method, which combines DNA models with embedding-based similarity search in vector databases. The advantages of our method are fourfold:

- **Mismatch tolerance:** Vector similarity based on our DNA models provides strong tolerance to mismatches, addressing the limitations of hash table–based string matching in handling sequence variation.
- **Higher specificity with longer seeds:** Longer seeds provide higher specificity within the genome, enabling the use of fewer seeds and thereby improving the efficiency of the seeding process.
- **Adaptive seed allocation:** Our method assigns fewer seeds to highly unique queries for efficiency and more seeds to complex queries for accurate localization.
- **GPU efficiency and scalability:** Optimized for GPU execution, the framework benefits from progress in GPU and vector database technologies, supporting rapid analysis in urgent clinical contexts such as infectious disease detection and intraoperative diagnosis.

## 2 PRELIMINARIES: WORKFLOW OF SEQUENCE ALIGNMENT

Given a DNA fragment, known as a *read*, of length $L_{read}$, DNA sequence alignment aims to determine its position on a reference genome of length $L_{genome}$. A direct yet the most accurate approach is to compare the read against every possible subsequence of the genome with the same length (Smith et al., 1981). However, this brute-force method has a computational complexity of $\mathcal{O}(L_{genome} \times L_{read})$. Since $L_{genome}$ can exceed 3 billion, as in the human genome, and $L_{read}$ typically ranges from 100 to 100,000, such an approach is computationally infeasible in practice.

To accelerate sequence matching, most alignment tools, including ours, adopt a heuristic *seed-chain-extend* strategy (Li & Durbin, 2009; Li, 2018). The strategy first identifies short exact or nearly exact matching substrings between the read and the genome, called *seeds*. These seeds then narrow the search space for more detailed alignment. Under this strategy, read mapping typically involves four steps: *Indexing* the reference genome, *Seeding*, *Chaining*, and *Alignment*. Below, we first introduce seed representation, which is central to fast indexing, followed by a description of these four steps.

**Seed Representation**: To accelerate seed matching, each seed $s$ is mapped into a new space $\mathbb{V}$ for efficient indexing and retrieval. Conventional methods define $\mathbb{V}$ as a fixed-length integer string space, while our method uses a vector space. Formally, the mapping $\mathcal{S}$ is defined as $\mathcal{S} : \{A, T, C, G\}^{L_{seed}} \rightarrow \mathbb{V}$, where $\{A, T, C, G\}$ are the four nucleobases (**A**denine, **T**hymine, **C**ytosine, **G**uanine), $L_{seed}$ is

the seed length. The representation of $s$ is obtained by $r = \mathcal{S}(s)$. For example, BWA-MEM (Li & Durbin, 2009) encodes substrings as base-4 integers, making $\mathcal{S}$ injective, while Minimap2 (Li, 2018) uses minimizers to select window-specific $k$-mers, minimizing redundancy and storage, while partially tolerating mismatches. However, in these conventional methods, $\mathbb{V}$ is not a metric space, so even a single nucleotide difference can lead to divergent representations. This makes their seeds highly sensitive to sequencing errors and genetic variations.

**Indexing**: Indexing is a mapping $\mathcal{I}$ that associates each substring with its genomic position(s), enabling efficient lookup during seeding. The genome is scanned using a sliding window of seed length, generating all substrings of that length. Their representations and positions are then stored in a specialized data structure, such as a hash table or vector database. Formally, indexing operates in the representation space and is defined as $\mathcal{I} : \mathbb{V} \rightarrow \mathbb{Z}^+$, with $\mathcal{I}(r) = y_j{}_{j=1}^m$ for a substring representation $r$, where $y_j$ is a matched position and $m$ is the number of matches. Since a substring may appear in multiple locations due to repeats or homology, the mapping can return several positions.

**Seeding**: Seeding identifies short subsequences with exact or near-exact matches between a query read and the reference genome. These matches yield candidate positions, called *anchors*, which serve as potential alignment start points. The process begins by extracting $n$ subsequences from the query read as seeds, denoted as $\{s_i\}_{i=1}^n$. For each seed $s_i$, we obtain its representation $r_i = \mathcal{S}(s_i)$ and retrieve its anchors $\{y_{ij}\}_{j=1}^{m_i}$. Equivalently, the anchors of a seed can be obtained through the composite mapping $\mathcal{I} \circ \mathcal{S}$.

**Chaining**: A single seed often matches multiple genomic locations, so one seed alone cannot determine the true genomic origin of a read. To resolve this, multiple seeds are extracted in a linear order from the read, and their corresponding anchors on the reference genome are grouped into chains. The goal is to find the longest co-linear chain of anchors that preserves the order and approximate spacing of seeds in the read. As shown in Fig. 1, this can be visualized as connecting anchors in a query–reference scatter plot with line segments of slope close to one (or minus one for the negative strand). The longest consistent trace is then selected as the candidate region for alignment. The problem of finding the longest consistent chain is usually solved with dynamic programming.

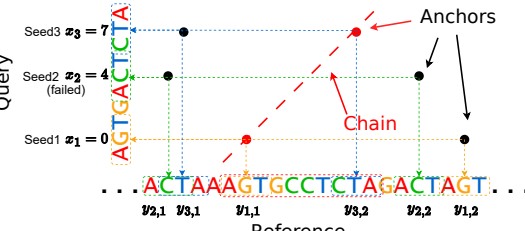

Figure 1: **Illustration of seeding and chaining.** Seeds $s_i$ are extracted from the query read at positions $x_i$, and their corresponding anchor positions $y_{i,j}$ are retrieved from an index of the reference genome. In the figure, each seed is connected to its anchors by colored dashed lines, forming a query–reference scatter plot. Chaining is then applied to this plot to identify the longest consistent sequence of anchors, which serves as the basis for downstream alignment.

**Alignment**: In the final stage, candidate regions from chaining are refined through nucleotide-level alignment, commonly using dynamic programming algorithms such as Smith–Waterman (Smith et al., 1981). This step fills in gaps between seeds and resolves substitutions, insertions, and deletions, producing a complete alignment represented by a CIGAR string (Li et al., 2009). To improve efficiency, banded Smith–Waterman is often used, restricting computation to the neighborhood of chained anchors and reducing the time complexity to $\mathcal{O}(L_{read} \times L_{gap})$, where $L_{gap}$ is the maximum distance between consecutive anchors.

## 3 METHODS

Most existing DNA alignment methods, including the SOTA Minimap2 (Li, 2018), rely on hash table–based string matching for fast indexing. Hash tables only support exact matches and cannot capture approximate similarity. In practice, sequencing errors and genetic variations often cause DNA reads to differ slightly from the reference genome. As a result, hash table indexing may fail to retrieve seeds that are highly similar but not identical.

To overcome this limitation, we propose representing seeds with neural network embeddings that preserve sequence similarity in a continuous space. In the following subsections, we first describe our neural encoder for DNA seeds; then introduce our genome indexing method based on embed-

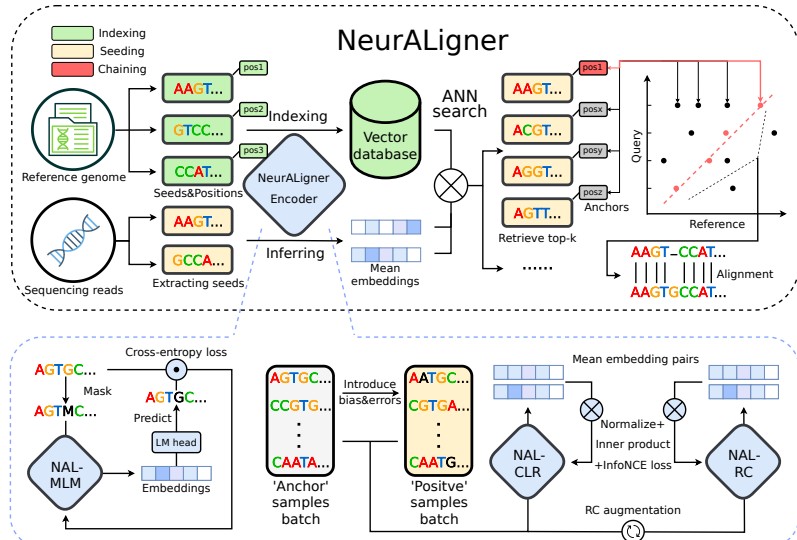

Figure 2: **Overview of the proposed NeurALigner**. The top figure demonstrates the process of aligning reads using NeurALigner, while the bottom figure illustrates the training of three NeurALigner Encoders.

dings and vector search, followed by our seeding and chaining procedure. Finally, we present an adaptive seed number strategy that balances efficiency and accuracy. The overall framework of our proposed method, **NeurALigner** (**NAL**), is illustrated in Fig. 2.

### 3.1 NEURAL ENCODER AND CONTRASTIVE TRAINING

The neural encoder is the central component of our alignment system. The quality of its generated embeddings directly affects alignment accuracy, while its inference speed influences both indexing and seeding efficiency.

**Training loss.** To obtain high-quality seed embeddings that are robust to sequencing errors, we train our model using self-supervised contrastive learning (Liu et al., 2021; Yu et al., 2023). Given a minibatch of $N$ sequences and their augmented counterparts, the encoder generates normalized representations $\mathbf{z}_i$ and $\mathbf{z}'_i$. Each sequence and its augmented version form a positive pair, while augmented versions of other sequences act as negatives. The model is optimized with the InfoNCE loss (Oord et al., 2018) with $\tau$ being a temperature parameter:

$$\mathcal{L}_{\mathrm{clr}} = -\frac{1}{N} \sum_{i=1}^{N} \log \frac{\exp(\mathbf{z}_i^{\mathrm{T}} \mathbf{z}'_i / \tau)}{\sum_{j=1}^{N} \exp(\mathbf{z}_i^{\mathrm{T}} \mathbf{z}'_j / \tau)}. \tag{1}$$

**Training data.** Training data are sampled from the Human Genome primary assembly of GRCh38.p14 (Nurk et al., 2022). We extract random fixed-length sequences from this reference genome, discard those containing the unknown base 'N,' and use the rest as training sequences.

Contrastive learning requires generating the augmented data as the positive sample for each training sequence. For each sequence, we construct its augmented version by introducing independent errors and shifts. Error rates $e_i$ are sampled from the uniform distribution $\mathcal{U}[0.01, 0.1]$, with substitutions, insertions, and deletions applied independently at each position, mimicking third-generation sequencing noise. After error injection, the sequence is shifted by $b_i$ bases, where $b_i$ follows a bounded normal distribution within $\pm L_{\mathrm{seed}}/10$ (with $L_{\mathrm{seed}} = 256$ by default). Sequences are then trimmed or padded to a fixed length.

Since the training objective of contrastive learning will push the training sequence and its positive sample to be close in the representation space, augmenting the sequence by introducing errors allows our model to extract embeddings robust to sequencing errors, while augmentation with shifting encourages translation continuity. Translation continuity is important in improving the efficiency and enhancing the robustness of alignment. It ensures that the embedding changes smoothly under small positional shifts. When an index stores embeddings only at a subset of positions, the true

match at query time may be offset by a few bases. With translation continuity, embeddings for nearby positions remain close in the representation space, so nearest neighbor search can still return the correct region despite slight misalignment. This property improves robustness and recall and enables a sparser index with lower memory and faster search. Without it, small shifts can cause large embedding changes, and retrieval becomes unreliable, which harms alignment quality.

**Ecnoder Architecture.** To ensure fast inference, the encoder must be compact and parallelizable. Although transformers (Vaswani et al., 2017) and recurrent networks (Yu et al., 2019) are effective for language and sequential data, they are not well-suited in the DNA alignment task, because transformers scale quadratically with sequence length, and recurrent networks lack efficient sequence-level parallelism. Instead, we adopt a convolutional architecture inspired by Hyena-DNA (Nguyen et al., 2023). We use its smallest variant, which has 0.5M parameters, enabling seed embeddings to be computed within microseconds.

Unlike Hyena-DNA, which is designed for sequence generation and trained with next-token prediction, our model is designed for embedding extraction and is trained with contrastive learning. This difference in objective requires two key modifications to the architecture. First, for contrastive training, we remove the language modeling head and take the final encoder output as the seed representation. To reduce the risk of dimensional collapse in contrastive learning (Jing et al., 2021), we add a trainable projection layer that maps embeddings before they are used for the loss function. Second, Hyena-DNA uses a unidirectional convolution kernel, which is effective for generation but less suitable for alignment. A unidirectional kernel restricts error propagation to one direction, which causes errors at the beginning of a seed to accumulate and become more harmful than those at the end. To address this issue, we redesign the convolution kernels in NAL Encoders to be bidirectional. With this design, errors occurring at different positions within a seed are treated almost equivalently.

### 3.1.1 ACHIEVING RC-INVARIANCE THROUGH DATA AUGMENTATION

In genomics, a reverse complement (RC) is functionally identical to the original sequence because of the complementary base-pairing in DNA, as illustrated in Fig. 7. In standard seeding, both a seed and its reverse complement must be indexed. If the embedding is reverse-complement equivariant, a single embedding can represent both orientations. This reduces seeding time by about half since only one computation and one query are needed per seed.

Prior works on RC-equivariant or RC-invariant DNA models primarily focus on architectural designs. For example, Shrikumar et al. (2017) designs RC-convolution layers with weight sharing, while Mallet & Vert (2021) provides a mathematical framework for designing RC-equivariant layers. These methods, however, require parameters to be reused multiple times during inference, which increases runtime and can outweigh the efficiency gains of having RC-invariant embeddings. More rcecent work (Schiff et al., 2024) enforces RC symmetry by mirroring the model and processing both the original and reverse complement sequences separately. While effective, this method may break translation continuity.

We adopt a simpler approach by encoding RC-equivariance through data augmentation. This strategy is widely used to encourage approximate invariance (Chen et al., 2020). During training, each sequence and its paired sequence are independently transformed into their reverse complements with a probability of 50%. As a result, neither, one, or both sequences may be complemented. The model is trained with the same contrastive objective as in Eq. (1). This RC data augmentation encourages the encoder to produce similar embeddings for a seed and its reverse complement. We refer to the model trained with RC-augmentation as **NAL-RC**, and the model trained without it as **NAL-CL**.

### 3.2 EMBEDDING-BASED GENOME INDEXING

To enable efficient retrieval of seeds with approximate matches, we represent subsequences of the genome using embeddings extracted by our NAL Encoder. Each subsequence has the same length as the seeds, and we store both its embedding and its starting position in a vector database. The robustness of these embeddings allows the retrieval process to tolerate sequencing errors.

We further take advantage of the translation continuity of the embeddings to build a sparse index for faster search. Instead of indexing every position in the genome with a sliding window of step size 1, we use a window of size $k$. Each seed embedding then represents $k$ consecutive bases, and for

each window, we index only the embedding of the seed centered at the middle base. This means that a seed from an error-free read can differ by up to $\lfloor \frac{k}{2} \rfloor$ bases from an indexed seed while their embeddings remain close in the embedding space. Combined with compression techniques in the vector database, this design achieves both efficiency and reduced memory usage. The index using vector database is implemented using Faiss (Johnson et al., 2019) as detailed in Appendix A.3.

## 3.3 SEEDING

The choice of seed length is an important trade-off in sequence alignment. Short seeds increase the number of seeds and generate more candidates, which slows down the alignment because more matches must be checked. Long seeds are more likely to be unique in the genome, which makes them more informative and reduces the number of seeds needed per read. However, sequencing errors often break exact matches for long seeds. In methods like Minimap2 that rely on hash table indexing, these errors can lead to failed lookups. To address this, Minimap2 favors short seeds, typically 15-19.

NeurALigner takes a different approach by embedding seeds into a continuous vector space. Each seed is represented by an embedding produced by our model, so sequences that are similar appear close in this space. A vector index with Approximate Nearest Neighbor search retrieves matching seeds efficiently, even when the query contains sequencing errors. This makes it possible to use much longer seeds, typically 256 or 512. With longer seeds, NeurALigner achieves both high recall and high precision while selecting only a small number of seeds per read.

Although a longer seed is more informative, it can still map to multiple locations in repetitive regions, yielding multiple anchors. While similarity scores from the index may indicate anchor reliability, they can be affected by index compression. Thus, a chaining phase to link consistent anchors is necessary to improve mapping accuracy.

## 3.4 CHAINING

In our method, both seeds and their anchors can be inexact. In other words, the positions of anchors may deviate slightly from the true positions of their corresponding seeds. To handle this, we relax the conventional chaining rules, loosening constraints on relative positions and accommodating larger gaps between neighboring anchors. This flexibility enables us to capture correct alignments even when errors cause anchors to shift. Formally, we identify the left-most genomic position of the best chain by solving the following problem:

$$y^+_{chain} = \arg \max_y [\sum_{i=1}^{n} \sum_{j=1}^{m} \mathbb{I}(|y_{i,j} - x_i - y| \leq C)],$$

$$y^-_{chain} = \arg \max_y [[\sum_{i=1}^{n} \sum_{j=1}^{m} \mathbb{I}(|y_{i,j} + x_i - L_{\text{read}} + L_{\text{seed}} - y| \leq C)].$$

(2)

Here, $y^+_{chain}$ and $y^-_{chain}$ are the left-most genomic positions of the best chain on the positive and negative strands. $y_{i,j}$ is the position of the $j$-th anchor for the $i$-th seed, $x_i$ is the position of the seed in the read, and $C$ is the allowed positional tolerance. For the negative strand, positions are adjusted to account for the reversed sequencing. A seed with relative position $x_i$ on a read whose left-most genomic position is $y$ has the actual position $y_{i,j} = y + L_{\text{read}} - L_{\text{seed}} - x_i$.

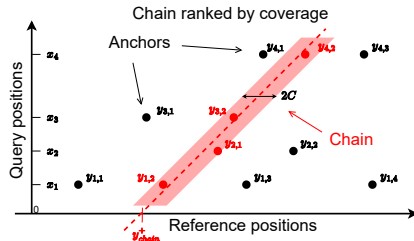

Figure 3: Illustration of Chaining in NeurALigner.

This chaining process can be interpreted as selecting a diagonal stripe on a query–reference scatter plot, as illustrated in Fig. 3. The stripe has slope 1 for the positive strand and slope –1 for the negative strand, with width $2C$. The optimal chain is defined as the one that contains the largest number of anchors rather than the longest span. In practice, we retain the top-$K$ chains with the most anchors for the next stage of alignment.

Table 1: Implementations of different phases in Minimap2, ESA, and NeurALigner

| Phase \Method | Minimap2 (Li, 2018) | ESA (Holur et al., 2025) | NAL (Ours) |
|---|---|---|---|
| Index type | hash table | vector database | vector database |
| Seed representation | minimizer | embedding | embedding |
| Seed length | $15 \sim 19$ | $L_{read}$ | $64 \sim 1024$ |
| Seed number | $\sim \lfloor L_{read}/10 \rfloor$ | 1 | $5 \sim 20$ |
| Chaining | ✓ | ✗ | ✓ |
| Alignment | Banded Smith-Waterman | Global Smith-Waterman | Wavefront algorithm |

While a single seed may not always be recalled, combining more seeds through chaining greatly improves the chance of correct alignment. This effect is captured in the following proposition, which states that alignment accuracy approaches 1 as the number of seeds tends to infinity.

**Proposition 1.** *Assume that embeddings from different read positions are independent, and that the recall $p_i$ for each seed $s_i$ under tolerance $C$ satisfies $p_i \geq p$ for some probability $p > 0.5$ and for all $i = 1, 2, ..., n$. Then the probability that the chaining position $y_{chain}$ is within $C$ of the ground-truth position $y_{gt}$ satisfies $P\{|y_{chain} - y_{gt}| \leq C\} \geq 1 - \exp\left(-\frac{(p-0.5)^2}{2p(1-p)}n\right)$.*

### 3.5 ADAPTIVE SEED NUMBER: RESCUE STRATEGY

If most seeds from a read produce anchors that align to the same genomic position, this position is likely the correct mapping location. If only a few seeds produce anchors that fall within the best chaining region, or if the top-$K$ chains have similar scores, where the score is the number of anchors contained in the chain, the mapping becomes less reliable.

To handle such cases, NeurALigner adopts a rescue strategy inspired by (Feng et al., 2014). In the first step, reads are seeded and chained using only a small set of seeds. For high-identity reads, i.e., reads that are very similar to the reference sequence, the best chain often achieves the maximum score, which equals the number of seeds sampled, and is directly accepted as the final mapping position. For reads with lower identity or those from repetitive regions, chains that fall below a score threshold or show ambiguous top-$K$ results are marked for further processing. In these situations, the rescue strategy adds more seeds and repeats the seeding and chaining process. This procedure continues until the read is either mapped with high confidence or discarded. In our implementation, we start with 5 seeds per read and increase to 16 seeds in the second round. Only one rescue round is usually needed, so we do not include a third. This adaptive allocation of seeds allows most reads to be mapped in the first round, while additional computation is reserved only for difficult cases.

### 3.6 ALIGNMENT

Our anchor points are approximate rather than exact matches. Therefore, the commonly used banded Smith–Waterman algorithm is not directly applicable to our NeurALigner, and running a full Smith–Waterman alignment would require $O(L_{\text{read}}^2)$ time, which is impractical for long reads. Instead, we adopt the wavefront algorithm (WFA) (Marco-Sola et al., 2021), which aligns sequences in $O(L_{\text{read}} \times s)$ time, where $s$ is the alignment penalty score. Unlike Smith–Waterman, WFA avoids computing full matching scores and focuses computation near the diagonal of the alignment matrix, making it highly efficient for reads with high sequence identity. We further accelerate alignment using a GPU-based WFA implementation (Aguado-Puig et al., 2023). WFA is most effective when the query and reference segments have similar lengths. To achieve this, for each candidate position, we extract from the reference a segment of length $1.002 \times L_{\text{read}}$ and use it as the alignment target.

**Remark: Comparison to Existing DNA Alignment Methods.** We have discussed the difference between our method and Minimap2 (Li, 2018). Here we focus on another related work, ESA (Holur et al., 2025), which is the first to apply DNA embeddings with vector database search for alignment. ESA encodes each short read as a single fixed-length embedding using a transformer model. For every read, it retrieves the top 50–75 most similar sequences from the database and applies Smith–Waterman alignment against each candidate. This design treats the entire read as a single seed and omits a chaining phase. ESA targets short reads of up to 500 bases but is about $100\times$ slower than BWA, which limits its use in practice. Still, ESA showed that deep models can represent DNA sequences effectively, which motivated our development of NeurALigner. The main differences between Minimap2, ESA, and NeurALigner (NAL) are summarized in Tab. 1.

Table 2: Comparison of mapping accuracy of models on reads with different identities

| Identity | NAL-CL | NAL-RC | NT | Hyena-DNA | Minimap2 |
|---|---|---|---|---|---|
| perfect | 99.997% | **100%** | Failed | 96.124% | **100%** |
| good | **100%** | **100%** | Failed | 93.757% | **100%** |
| normal | **99.643%** | 99.600% | Failed | 78.306% | 99.483% |

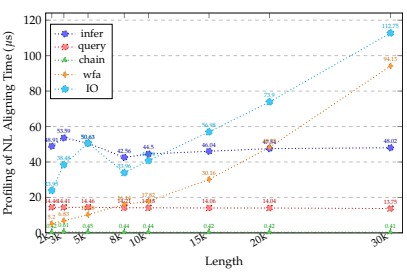

Figure 4: Runtime of each component across seed lengths on reads with 90% identity.

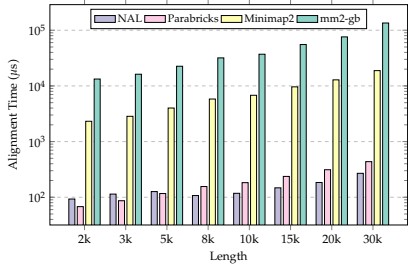

Figure 5: Runtime comparison across seed lengths on reads with 90% identity.

# 4 EXPERIMENTAL STUDIES

To evaluate the effectiveness of NeurALigner, we design experiments to answer three questions. **Q1**: Can the embeddings produced by the encoder allow accurate alignment, even when the reads have low identity? **Q2**: How efficient is NeurALigner in performing alignments? **Q3**: Can a neural encoder trained on the human genome be applied to alignment tasks in other species? Additional results are provided in Appendix B. We use Minimap2 (Li, 2018) as our primary baseline due to its SOTA accuracy, efficiency, and widespread adoption in practice. We exclude ESA from comparison since it is tailored to short reads and is not applicable to the long reads in our experiments.

**Preparing Reads for Alignment** Accurate ground truth mapping for real sequencing data is not available because the true genomic origin of each read cannot be measured at scale. To enable reliable benchmarking, we generate synthetic reads by extracting segments from the reference genome, recording their true origin, and adding error patterns that reflect different sequencing technologies. This setup gives us reads with controlled length, identity, and documented origin.

We simulate reads at three identity levels to the reference. The *perfect* set has an average of 99.9% identity and a minimum of 99%. The *good* set has an average of 98% and a minimum of 95%. The *normal* set has an average of 95% and a minimum of 85%, modeling standard Nanopore-2023 reads. To study the impact of multimapping on alignment accuracy, we use two types of origin regions in the human genome (Nurk et al., 2022): (1) a 10 Mbp segment, chr1:170M–180M, deliberately chosen to avoid ENCODE blacklist regions (Amemiya et al., 2019), and (2) the entire human genome, which represents standard alignment conditions. See Appendix A.1 for more details.

## 4.1 ACCURACY OF NEURALIGNER EMBEDDINGS

To test whether the embeddings from NeurALigner enable accurate alignment, we compared our models with Minimap2 on reads of different identity levels. Here, A read is considered correctly mapped if its mapped target window overlaps at least 95% with the read's true position on the reference genome. We report results for two versions of NeurALigner. NAL-CL is trained with contrastive learning, while NAL-RC adds reverse complement augmentation. Tab. 2 shows the mapping accuracy on simulated reads. At high identity, both NAL and Minimap2 reach nearly perfect accuracy. At the normal identity level, where sequencing errors are more frequent, NAL slightly outperforms Minimap2. These results indicate that NAL achieves the same level of accuracy as Minimap2, which demonstrates that it is equally applicable in practical scenarios. In Sec. 4.2, we will further show that NAL provides a clear advantage in efficiency compared with Minimap2.

We also test other DNA model architectures, including Nucleotide Transformer (NT) (Dalla-Torre et al., 2025) and Hyena-DNA (Nguyen et al., 2023), by substituting our NAL Encoders with them, keeping all other configurations unchanged. As shown in Tab. 2, NT fails in all cases, which is consistent with the findings in (Holur et al., 2025). Additionally, Hyena-DNA performs much worse

than NAL-CL, with accuracy dropping sharply as identity decreases. These results confirm that our architectural design and training strategy are important for achieving robust alignment.

## 4.2 EFFICIENCY OF NEURALIGNER

NAL achieves high efficiency by reducing the cost of seeding and by taking advantage of vectorization. Minimap2 extracts seeds at every position, which requires $\mathcal{O}(L_{\text{read}})$ operations. In contrast, NAL needs a few informative seeds. With the rescue strategy, an average of 5-8 seeds per read is sufficient even when read length is up to 30k bases. This is especially beneficial for long reads. Fig. 4 confirms that the costs of seeding and chaining remain constant, while the cost of alignment and I/O increases with read length. The WFA is efficient for high-identity reads ( Fig. 8 in Appendix), but decreases for these noisier reads. Tab. 7 in Appendix reports similar results for high-identity reads, where NAL uses only 5.39 seeds per read. The main bottlenecks are I/O and SAM output.

Fig. 5 compares the alignment speeds of NAL with Minimap2 and its two GPU variants, Parabricks (NVIDIA, 2025) and mm2-gb (Dong et al., 2024). We test reads of different lengths, using 5,000 reads per length and duplicating them 300 times to ensure stable timing. GPU baselines run on a single NVIDIA H20 GPU with a 256-core CPU server, while Minimap2 runs on CPU. NAL achieves the fastest runtimes, up to 276× faster than Minimap2 on 30k-base reads. Although NAL is slower than Parabricks on short reads under 5k, its advantage grows with read length, surpassing Parabricks beyond 8k. The time to load indices into GPU, which is less than 1 second, is excluded.

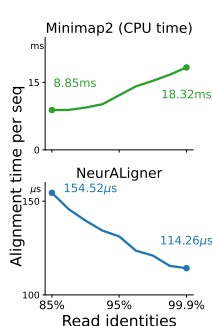

Figure 6: Runtime comparison across read identity levels

To further demonstrate the advantage of efficient seeding, Fig. 6 compares alignment time across read identities. NAL maintains low latency, while Minimap2 slows down by up to 2× on high-identity reads because redundant seeds add chaining overhead. NAL avoids this by using longer seeds with higher recall, and adopts the rescue strategy to keep it efficient.

## 4.3 GENERALIZATION TO OTHER SPECIES

A key question is whether NeurALigner learns generalizable nucleotide embeddings or only memorizes the human genome. To test this, we use the encoder trained on human data to evaluate alignment accuracy on simulated reads from mouse (GRCm39, *Mus musculus* (Waterston et al., 2002)) and zebrafish (GRCz11, *Danio rerio* (Howe et al., 2013)). The details of data generation are available in Appendix A.1. Tab. 3 shows that NAL achieves high accuracy across both species. For mouse reads,

Table 3: Alignment accuracy on mouse and zebrafish.

| Species | Identity | NAL-CL | NAL-RC | Minimap2 |
|---------|----------|--------|--------|----------|
| Mouse | Perfect | 99.516% | 99.252% | 100% |
| | Good | 99.505% | 99.249% | 99.997% |
| | Normal | 99.038% | 98.557% | 99.498% |
| Zebrafish | Perfect | 99.950% | 99.952% | 99.996% |
| | Good | 99.959% | 99.952% | 99.996% |
| | Normal | 99.548% | 99.505% | 99.471% |

the accuracy remains close to that of Minimap2. For zebrafish, NAL even surpasses Minimap2 on reads with normal identity, without training on the zebrafish genome. These results suggest that the encoder captures sequence features that extend beyond the training genome, enabling robust alignment in different species.

## 5 DISCUSSION

We presented NeurALigner, a GPU-accelerated alignment framework that reformulates seed matching as embedding-based similarity search. By combining robust DNA embeddings, adaptive seeding, and efficient chaining with vector databases, NeurALigner achieves both high accuracy under sequencing noise and substantial speedups over existing aligners. Experiments show its ability to generalize across species and maintain performance on long reads, making it suitable for large-scale and time-sensitive genomic analysis. Looking ahead, we see opportunities to extend the framework with more advanced DNA foundation models, integration with downstream variant calling, and further optimization for distributed GPU clusters.

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

# Appendix

## Table of Contents

## A  SUPPLEMENTARY EXPERIMENT DETAILS

### A.1  DATA GENERATION FOR EVALUATION

We use Badread (Wick, 2019) to simulate Oxford Nanopore (Branton et al., 2008) reads. Badread provides both error modeling and QScore modeling, and we adopt the error model in our experiments. In this model, errors are introduced by selecting $k$-mers in a sequence and replacing them with new $k$-mers, with occasional reversions to the original. Error probabilities are determined by a transfer matrix, which allows Badread to reproduce substitution patterns typical of different sequencing technologies. In addition to substitutions, Badread can also generate random reads, glitches such as overlapping segments or long deletions, and chimeric reads formed by joining unrelated fragments.

To evaluate alignment speed on reads of fixed length, we also construct synthetic sequences by introducing substitutions, insertions, or deletions at each base with fixed probabilities. Although these synthetic reads do not fully match real sequencing data, they provide a controlled setting to demonstrate the speed of our method on long reads.

All reads from the negative strand are converted to their reverse complement so that the initial dataset contains only positive-strand reads. For dual-strand alignment experiments, half of the reads are again converted to their reverse complement. Each read is labeled with its true leftmost genomic position to enable accuracy evaluation. Pure noise sequences are not labeled, and they are considered correctly aligned if they produce no matches.

For each identity level, we generate four sets of reads: a(1) ∼965,000 reads from the entire human genome, (2) ∼32,000 reads from human chr1:170M–180M, (3) ∼32,000 reads from the mouse genome, and (4) ∼23,000 reads from the zebrafish genome. These datasets are generated using Badread, with read lengths between 1 kb and 32 kb. The average length is 15 kb. Reads longer than 32 kb are excluded.

For fixed-length read benchmarks, we manually generate 5,000 human reads for each of eight lengths (2 kb, 3 kb, 5 kb, 8 kb, 10 kb, 15 kb, 20 kb, and 30 kb) and four identity levels (90%, 95%, 99%, and 99.9%). These reads are duplicated several times to obtain stable measurements of alignment speed.

### A.2 TRAINING DETAILS

The model takes strings as input, where each character is automatically tokenized into its ASCII code. We adopt the same architecture as Hyena-DNA tiny-1k, but remove the language modeling head, add a trainable projection for contrastive learning, and replace the unidirectional convolution kernels with bidirectional ones. The output of the model has the shape $[B, L, D]$, where $B$ is the batch size, $L$ is the input length, and $D$ is the embedding dimension. The embedding of a sequence is obtained by averaging along the $L$ dimension, after which the embeddings are normalized to unit $L_2$ norm. In our implementation, $L$ ranges from 64 to 1024, and $D = 128$.

All models are optimized with the AdamW optimizer (Loshchilov & Hutter, 2017). The learning rate is set to $1 \times 10^{-3}$, the weight decay is 0.1, and the coefficients $\beta_1$ and $\beta_2$ are 0.9 and 0.999. The learning rate is reduced by 1/5 if the validation loss does not decrease for four consecutive epochs. Training stops if the validation loss does not improve for twelve epochs.

To increase diversity, training data are not pre-generated. Instead, new samples are created at every step. The training and validation sets are drawn from the same distribution but with different random seeds to keep them independent. The random seed is fixed so that models of the same type and sequence length see identical data at each step.

For NAL-CL and NAL-RC, the InfoNCE minibatch size is 8,192, with 128 steps per epoch. The best performance is reached after processing about 13 million sample pairs.

Training a single model requires about 30 minutes on eight NVIDIA H20 GPUs, using PyTorch and PyTorch Lightning.

### A.3 INDEXING CONFIGURATION

We build the index using Faiss (Johnson et al., 2019), with inverted file and product quantization (IVFPQ). IVFPQ reduces the cost of approximate nearest neighbor search to about $\mathcal{O}(\sqrt{N})$ by clustering the vectors and restricting comparisons to cluster centroids and their neighbors. While the Hierarchical Navigable Small World (HNSW) index can also be fast, it requires much more memory, which makes it less practical in our setting.

All vector similarity searches are performed on the GPU to fully exploit parallelism. This design makes the entire seeding phase GPU-accelerated, but it also requires the index to fit into GPU memory. Product quantization (Jegou et al., 2010) addresses this constraint by reducing the storage size of embeddings. For instance, a 128-dimensional FP32 embedding can be compressed to 24 bytes using the PQ16x8 setting in Faiss. With product quantization and an indexing stride of 32, the GRCh38.p14 primary assembly requires only 2.1 GB of GPU memory, which is feasible on most modern GPUs.

We follow a simple set of guidelines for configuring the index. The number of clusters `nlist` is set to the square root of the total number of vectors. The indexing stride should be no larger than 1/8 of the seed length, since larger strides cause a notable loss of accuracy. For the vector search, the number of clusters to probe `nprobe` is set between 8 and 32. Probing more clusters yields only small gains in recall but slows down search considerably. For each seed, we keep the top $K$ nearest neighbors, which serve as candidate anchors in the chaining phase. By default, $K = 32$. Additionally, we use the inner product as the distance metric.

Indexing the entire human genome on a single NVIDIA H20 GPU takes between 20 and 50 minutes, depending on the configuration.

### A.4 IMPLEMENTATION OF CHAINING

The chaining algorithm has a time complexity of $\mathcal{O}(N \cdot MK \log(MK))$, which is dominated by the sorting step. Since $M$ and $K$ are bounded by constants independent of the read length $L_{read}$, the overall complexity does not grow with $L_{read}$. The results of this algorithm may differ slightly from those of Eq. 2, because the stripe axis is fixed to anchors with the locally maximum retrieval similarity, and anchors within $C_b$ are merged into the stripe. All operations in the pseudocode can be efficiently vectorized using NumPy (Harris et al., 2020) and PyTorch.

---

**Algorithm 1 Vectorized chaining** for indexes that use distance as the metric

---

**Require:** Read count $N$, number of seeds per read $M$, number of anchors per seed $K$, bias tolerance $C_b$, score threshold $n_0$, relative seed positions $\mathbf{X} = [x_{ij}] \in \mathbb{Z}^{N \times M}$, anchor positions $\mathbf{Y} = [y_{ijk}] \in \mathbb{Z}^{N \times M \times K}$, retrieved similarities $\mathbf{D} = [d_{ijk}] \in \mathbb{R}^{N \times M \times K}$

1: Set anchor scores $\mathbf{C} = [c_{i,jk}] \in \mathbb{Z}^{N \times MK}$, with $c_{i,jk} = 1$
2: Expand $\mathbf{X}' = [\mathbf{X}'_k]$, $\mathbf{X}'_k = \mathbf{X_k}$, $k = 1, \cdots, K$
3: Adjust $\mathbf{Y} = \mathbf{Y} - \mathbf{X}'$. For the negative strand set $\mathbf{Y} = \mathbf{Y} + \mathbf{X}' + L_{read} - L_{seed}$
4: Reshape $\mathbf{Y}, \mathbf{D}$ to $\mathbb{Z}^{N \times MK}$
5: For each read in parallel compute sorted index $\mathbf{I}_1 = \text{argsort}(\mathbf{Y})$ in ascending order
6: Rearrange $\mathbf{Y} = \mathbf{Y}[\mathbf{I}_1]$, $\mathbf{D} = \mathbf{D}[\mathbf{I}_1]$
7: **for** $j = 0$ **to** $MK - 2$ **do**
8:     In parallel across reads compute $\mathbf{d}_{max} = \max(\mathbf{D}_{:,j}, \mathbf{D}_{:,j+1})$ and $\mathbf{y}_{max} = \arg\max(\mathbf{D}_{:,j}, \mathbf{D}_{:,j+1})$
9:     In parallel across reads compute merge index $\mathbf{I}_2^j = (\mathbf{Y}_{:,j+1} - \mathbf{Y}_{:,j} \le C_b)$
10:     Update $\mathbf{D}_{:,j+1}[\mathbf{I}_2^j] = \mathbf{d}_{max}$, $\mathbf{Y}_{:,j+1}[\mathbf{I}_2^j] = \mathbf{y}_{max}$, and $\mathbf{C}_{:,j+1}[\mathbf{I}_2^j] = \mathbf{C}_{:,j+1}[\mathbf{I}_2^j] + \mathbf{C}_{:,j}[\mathbf{I}_2^j]$
11:     Set $\mathbf{D}_{:,j}[\mathbf{I}_2^j] = -\inf$, $\mathbf{Y}_{:,j}[\mathbf{I}_2^j] = -1$, $\mathbf{C}_{:,j}[\mathbf{I}_2^j] = -1$
12: **end for**
13: For each read in parallel compute sorted index $\mathbf{I}_3 = \text{argsort}(\mathbf{C})$ in descending order
14: Rearrange $\mathbf{Y} = \mathbf{Y}[\mathbf{I}_3]$, $\mathbf{D} = \mathbf{D}[\mathbf{I}_3]$, $\mathbf{C} = \mathbf{C}[\mathbf{I}_3]$
15: For each read in parallel compute acceptance index $\mathbf{I}_4 = (\mathbf{C}_{:,0} \ge n_0)$
16: **return** accepted chain positions $\mathbf{Y}_{:,0}[\mathbf{I}_4]$ and indices of rejected reads where $\mathbf{I}_4$ is False

---

### A.5 ASSESSING MODELS AND INDEX

From each read, we uniformly select 5 seeds. Their embeddings are computed, and the top 256 nearest neighbours for each embedding are retrieved. Recall@k is measured by checking whether a correct retrieval is present among the top-$k$ results. All indices used in these experiments are built with the configuration `IVF16384,PQ16`, stride 16, and query parameter `nprobe` set to 32.

For alignment accuracy experiments, each dataset is aligned using indices built on the complete genome of the corresponding species. The indexing configuration is `IVF16384,PQ16`, with `stride` set to 16 and `nprobe` set to 8. We use 7 seeds in the first iteration and 13 seeds in the second, and set top-$k$ to 32. These settings are chosen to maximize alignment accuracy. The alignment phase is omitted, since our chaining stage already selects one or more windows covering the true genomic locus of each read. A read is considered correctly mapped if at least one mapped window overlapped 95 percent or more of its true position. Random reads are considered correctly mapped only if no window is mapped. These same accuracy criteria are applied to Minimap2 for fair comparison.

For speed experiments, we use the fixed-length and identity dataset. Indexes are identical to those in the accuracy experiments, except that the number of seeds per iteration is set to 5 and 11. To avoid warm-up effects, each read is duplicated 300 times to create a large dataset, as PyTorch exhibits noticeable lag before the first inference. We measure the runtime of each phase of NeurALigner after the alignment process completes. For comparison, we also record the alignment time of WFA-GPU and the CPU runtime of Minimap2. The input and output time reported in Fig. 4 may fluctuate, since Python is inefficient at loading FASTA files and printing SAM output.

All evaluation experiments for our models and indexing are conducted on a single NVIDIA H20 GPU.

## B SUPPLEMENTARY EXPERIMENTAL STUDIES

### B.1 EFFECT OF SEED LENGTH

We examine whether longer seeds provide higher specificity in the NAL framework by measuring recall for single seeds of different lengths using NAL-CL. All indexes are built with the same configuration, IVF16384 with PQ16 and a stride of 16. A retrieval is considered correct if the difference between the predicted and true positions is no more than 64 bases. The results in Tab. 4 show that recall at rank 1 increases steadily as the seed length grows, across all three identity levels. This demonstrates that longer seeds provide more informative signals for alignment.

Table 4: Recall@1 with different seed length

| identity$\backslash L_{seed}$ | 128 | 256 | 384 | 512 |
|---|---|---|---|---|
| perfect | 90.08% | 94.16% | 95.21% | 95.57% |
| good | 86.99% | 92.61% | 94.25% | 94.83% |
| normal | 75.50% | 83.93% | 86.74% | 88.41% |

Table 5: Alignment accuracy for reads sampled from the whole genome.

| Identity | NAL-CL | NAL-CL | NAL-RC | NAL-RC |
|---|---|---|---|---|
| $L_{seed}$ | 256 | 512 | 256 | 512 |
| perfect | 99.446% | 99.507% | 99.620% | 99.710% |
| good | 99.398% | 99.460% | 99.320% | 99.481% |
| normal | 98.891% | 98.985% | 98.940% | 99.100% |

## B.2 EVALUATION WITH READS FROM THE WHOLE GENOME

In previous experiments, query reads are drawn only from non-repetitive regions of the genome. To create a more difficult but more real test, we now sample query reads from the entire genome, which includes many repetitive sequences and regions of homology. These characteristics introduce frequent cases of multimapping, making alignment more challenging.

As shown in Tab. B.2, both NAL-CL and NAL-RC models show a consistent decrease in alignment accuracy compared with the results on non-repetitive reads. Compared to the results shown in Tab. 2, the drop is about 0.5% across different identity levels when the seed length is 256.

Increasing the seed length improves the seed specificity within the genome and, therefore, reduces the occurrence of spurious matches. It provides a useful way to handle repetitive regions in whole-genome alignment. As observed, it gave an improvement in accuracy of about 0.05% to 0.1%.

## B.3 RC-INVARIANT MODEL ALIGNS READS FROM BOTH STRANDS SIMULTANEOUSLY

So far, we have considered alignment only for reads originating from the positive strand of the reference genome. In practice, however, reads can come from either strand, and the strand orientation is usually unknown before alignment. Minimap2 addresses this issue by aligning each read in both its original and reverse-complement orientations, then selecting the alignment with the higher score (Li, 2018). While effective, this strategy doubles the computational cost.

The NAL-RC model avoids this overhead by using reverse-complement invariance. It produces nearly identical embeddings for a sequence and its reverse complement, with an average cosine similarity of 0.9995 and a minimum of 0.99 in our experiments. As a result, a single mean embedding for each seed can represent both strands in the index. During seeding, NAL-RC retrieves anchors for both strands simultaneously. Strand orientation is then resolved during chaining, which is performed twice. Because chaining is highly efficient in our vectorized implementation, NAL-RC completes alignment for both strands with almost no additional computational cost.

The alignment accuracy of NAL-RC is comparable to that of NAL-CL. Although NAL-RC shows slightly lower performance during training, it maintains high accuracy in alignment, achieving 100% for reads of perfect and good identity, and 99.600% for reads of normal identity.

Table 6: Recall of a single seed and correlation check.

|  |  | NAL-CL (256) | NAL-CL (512) | NAL-RC (256) | NAL-RC (512) |
|---|---|---|---|---|---|
| Perfect | Recall@1 | 94.16% | 95.57% | 94.41% | 96.35% |
|  | Recall@32 | 97.63% | 98.14% | 97.58% | 98.53% |
|  | MRR | 0.9519±0.1990 | 0.9630±0.1764 | 0.9534±0.1968 | 0.9697±0.1600 |
| Good | Recall@1 | 92.61% | 94.83% | 92.52% | 95.51% |
|  | Recall@32 | 97.16% | 97.89% | 96.97% | 98.17% |
|  | MRR | 0.9396±0.2204 | 0.9569±0.1895 | 0.9384±0.2233 | 0.9627±0.1767 |
| Normal | Recall@1 | 83.93% | 88.41% | 81.91% | 88.24% |
|  | Recall@32 | 92.12% | 94.61% | 90.70% | 94.06% |
|  | MRR | 0.8632±0.3218 | 0.9016±0.2792 | 0.8446±0.3402 | 0.8990±0.2838 |
| Correlation |  | 0.0239 | 0.0262 | 0.0231 | 0.0214 |

# C  PROOF OF PROPOSITION 1

*Proof.*

$$P\{|y_{\text{chain}} - y_{\text{gt}}| \leq C\}$$

$$= P\left\{ \left| \arg\max_y [\sum_{i=1}^{n}\sum_{j=1}^{m} \mathbb{I}(|y_{i,j} - x_i - y| \leq C)] - y_{\text{gt}} \right| \leq C \right\}$$

$$\geq P\left\{ \sum_{i=1}^{n} r_i > \lfloor n/2 \rfloor \right\} \tag{3}$$

$$\geq \sum_{i=\lfloor n/2 \rfloor + 1}^{n} \binom{n}{i} p^i (1-p)^{n-i}$$

$$\geq 1 - \exp(-an) \to 1$$

$$\text{where} \quad r_i = \begin{cases} 1, \ \exists j, |y_{i,j} - y_{\text{gt}}| \leq C \ \text{(a successful retrieval)} \\ 0, \ \text{otherwise} \end{cases}, \quad a = \frac{(p-0.5)^2}{2p(1-p)}$$

The inequality third line holds because the chain will always be retained when the chain contains more than half anchors. □

To check the assumptions of Proposition 1, we evaluated two aspects.

First, we examined whether the recall of each seed is larger than 0.5. Tab. 6 reports Recall@1, Recall@32, and Mean Reciprocal Rank (MRR, (Craswell, 2016)) at three identity levels. The mean reciprocal rank (MRR) is calculated as the average reciprocal of the rank of the first correct retrieval, with a value of zero assigned when no correct match is found. We also report the standard deviation of the MRR. Across all settings, the recall values are well above 50%, confirming that the condition $p_i \geq 0.5$ is satisfied.

Second, we measured whether embeddings from different read positions are independent. We computed the correlation matrix between the embeddings of two random non-overlapping seeds in each read. Specifically, given the embeddings, calculate the Pearson correlation coefficient across all pairs of the 128 dimensions, and obtain a $128 \times 128$ correlation matrix. The maximum absolute correlation value from this matrix is reported as "Correlation" in Tab. 6 to evaluate redundancy among embedding dimensions. The correlation values in all experiments are below 0.03, which shows that embeddings produced by NAL Encoders are essentially uncorrelated.

Together, these results support the independence and recall assumptions in Proposition 1.

Table 7: Average runtime breakdown for reads of length 15,000 bases with 99.9% identity. In this case, NAL uses only 5.39 seeds per read on average. We can observe that seeding and chaining costs remain constant across both length and identity, while alignment and I/O vary. Therefore, the current bottlenecks are load reads and SAM output.

| Phase | Time |
|---|---|
| Load | $35.8\mu s$ |
| Seed(Infer) | $7.8\mu s \times 5.39$ |
| Seed(Query) | $2.5\mu s \times 5.39$ |
| Chain | $0.43\mu s$ |
| Align | $6.0\mu s$ |
| Print SAM | $16.4\mu s$ |
| Total | $114.3\mu s$ |

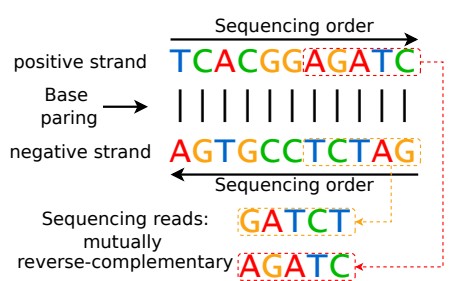

Figure 7: Illustration of DNA base pairing and RC reads

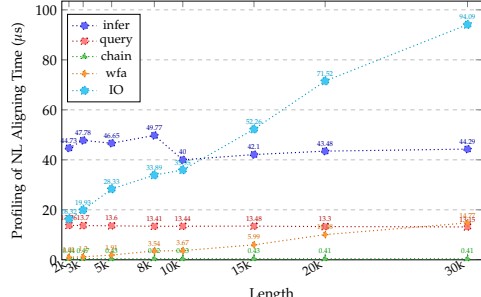

Figure 8: Runtime of each component vs. seed length for reads with 99.9% identity

## D  USE OF LLMS

We use LLMs to assist with refining the writing.

