# OpenReview forum: "NeurALigner: Scalable and Robust DNA Sequence Alignment via Embedding-Based Similarity Search"
_ICLR.cc/2026/Conference — ICLR 2026 Conference Withdrawn Submission_

### Official Review · Reviewer_iAp2 · 2025-10-18

**Soundness:** 3
**Presentation:** 4
**Contribution:** 2
**Rating:** 4
**Confidence:** 4

**Summary:**

This manuscript presents a refined approach to the seed-chain-align workflow for DNA sequence alignment, a fundamental problem in bioinformatics. The authors introduce a neural encoder, trained using a contrastive learning objective, to generate robust embeddings that preserve similarity for augmented DNA sequences. This allows the seeding step to be performed as a fast, approximate similarity search, making the process tolerant to sequencing errors and variations. While achieving accuracy comparable to the conventional baseline Minimap2, the proposed method significantly reduces runtime by leveraging GPU acceleration.

**Strengths:**

+ The application of neural embeddings for seed-based alignment.
+ A substantial improvement in performance, achieving significant time savings over conventional methods.
+ The manuscript itself is exceptionally clear and well-organized, making it easy to understand.

**Weaknesses:**

The reviewer considers the work's main innovation to be the neural encoder, which represents seeds via neural network embeddings to preserve sequence similarity in a continuous space, and the subsequent application of these embeddings in the seeding stage.

However, the concept of creating similarity embeddings for DNA sequences with neural networks is not novel, and the authors' survey of related work appears to be incomplete.

 + The following references and related works have already explored neural network encodings for DNA sequences, where the distance\similarity between embedding vectors is designed to approximate the distance\similarity between the sequences themselves.
+ These methods were developed for applications like fast clustering and sequence alignment [3,4]. Consequently, the core idea of using an embedding space for efficient similarity evaluation is not eye-catching novel, as they are just designed for this purpose.
 + Furthermore, the use of contrastive or metric learning with positive and negative samples has also been previously explored. For instance, reference [2] employs a triplet loss, and [4] utilizes a Siamese neural network.

Considering this, the application of existing neural network methods to sequence alignment represents a seemingly incremental novelty, especially in the context of ICLR rather than a bioinformatics conference.

1. S Koide, K Kawano, and T Kutsuna. Neural edit operations for biological sequences. NeurIPS 2018
2. X Zhang, Y Yuan, and P Indyk. Neural embeddings for nearest neighbor search under edit distance
3. X Dai, X Yan, K Zhou, Y Wang, H Yang, and J Cheng. Convolutional embedding for edit distance. SIGIR 2020
4. A Guo, C Liang, and Q Hou. Deep squared Euclidean approximation to the Levenshtein distance for DNA storage. ICML2022

**Questions:**

+ Please address the questions raised in the weakness section.
+ The proposed method uses a data-driven encoding model, which the authors claim generalizes from human genomes to mouse and zebrafish genomes. Would training such a model on randomly generated samples and augmentations also allow it to perform well in general applications?
+ Based on the manuscript's description, the goal of the RC augmentation seems to be achieving RC-invariance, rather than RC-equivariance. The authors may clarify their use of this terminology.

---

### Official Review · Reviewer_BemB · 2025-10-26

**Soundness:** 2
**Presentation:** 3
**Contribution:** 2
**Rating:** 4
**Confidence:** 3

**Summary:**

The paper introduces NeurALigner (NAL), a GPU-accelerated DNA sequence alignment framework that combines DNA sequence models with vector database retrieval. Departing from conventional exact string matching, the framework encodes DNA subsequences into embedding vectors and conducts similarity searches in the feature space, which yields significant gains in both efficiency and performance. The paper claims NAL achieves scalable, mismatch-tolerant alignment with high specificity and strong GPU performance.

**Strengths:**

**Originality**: This paper introduces modifications to the Hyena-DNA model architecture and employs contrastive learning to train the model for deriving DNA sequence embeddings, demonstrating a certain degree of innovation.

**Quality**: The paper primarily employs comparative experiments with Minimap2 to demonstrate that NeurALigner achieves comparable or superior performance in scenarios with sequence alignment and other challenges.

**Clarity**: The paper is well-structured and clearly explains the methodology, including the neural encoder, genome indexing, seeding, and chaining procedures.

**Weaknesses:**

- The fonts in the figures  are inconsistent and should be standardized.

- The claim regarding performance is undermined by a lack of comprehensive benchmarking, with comparisons drawn almost exclusively against Minimap2.

**Questions:**

- Given that a minor change in DNA can lead to different amino acids and even altered proteins, is research on approximate DNA sequence matching worth pursuing?

- The paper treats Minimap2 as the SOTA baseline, but it is from 2018; the authors should address whether any work in the same application domain has been published over the past seven years.

---

### Official Review · Reviewer_sACC · 2025-10-28

**Soundness:** 1
**Presentation:** 3
**Contribution:** 2
**Rating:** 2
**Confidence:** 3

**Summary:**

The paper proposes NeurALigner (NAL), a GPU-accelerated DNA sequence alignment framework. Unlike traditional seed-chain-align methods that rely on exact k-mer matches, NAL encodes DNA subsequences (seeds) into embeddings using a neural network inspired by Hyena-DNA but modified for bidirectional processing and trained with contrastive learning. Seed matching is reformulated as an approximate nearest neighbor search in a vector database built from reference genome embeddings. The framework uses longer seeds (e.g., 256bp) enabled by the embedding's mismatch tolerance, an adaptive seeding strategy, and a modified chaining algorithm tolerant to inexact anchor positions. The final alignment uses the wavefront algorithm (WFA). The authors claim NAL achieves accuracy comparable or superior to Minimap2, especially on noisy reads, while being significantly faster (claiming up to 276x speedup) due to GPU acceleration and efficient seeding. They also present a variant (NAL-RC) using data augmentation for reverse-complement invariance.

**Strengths:**

Generally, I like the paper's approach and it does show an interesting and promising direciton for future research in this area. More specifically:

* The main idea of using learned (contrastive) embeddings within a seed-chain-align framework to achieve mismatch tolerance during seeding is intuitively appealing and potentially valuable for future research
* The paper demonstrates empirically that the NAL framework achieves accuracy comparable to the state-of-the-art Minimap2 aligner,  showing a potential slight advantage on reads with lower identity
* The modification of a Hyena-DNA-like architecture to use bidirectional kernels and contrastive training seems appropriate for the goal of the paper, same for the data augmentation for reverse complement learning which makes sense
* The framework is designed for GPU acceleration sounds promising

**Weaknesses:**

## Theory
The paper's only formal theoretical result, Proposition 1, aims to guarantee the chaining strategy's accuracy scaling. However, its supporting argument is fundamentally flawed as elaborated below:
* The proof crucially assumes that the recall probability ($p_i$) for each individual seed is greater than 0.5 ($p_i > 0.5$). The only justification provided is an empirical appeal to high Mean Reciprocal Rank (MRR) values (>0.8) reported in Table 6. This justification is invalid because it fundamentally misinterprets the MRR metric: MRR measures the average reciprocal rank of the first correct item retrieved across many queries. A high MRR indicates that *on average* the first correct item is found quickly (often at rank 1), but it provides no mathematical guarantee whatsoever nor a lower bound on the probability ($p_i$) that a correct item will be retrieved within the top-K results for a *specific* seed. In other words, it's entirely possible to achieve high MRR even if a significant fraction of seeds have $p_i = 0$. Therefore, the central premise ($p_i > 0.5$) required by the proof is not unsubstantiated.
 * Besides the invalid premise, the proof makes a heuristic leap by assuming that recalling correct anchors for a majority ($>n/2$) of seeds guarantees that the chaining algorithm (which maximizes anchor count) will identify the correct genomic location ($y_{gt}$):

> $P\\{|y_{chain} - y_{gt}| \le C\\} \ge P\\{\sum r_i > n/2\\}$

 this step lacks formal rigor and does not account for potential failure modes like repeats causing incorrect chains to have slightly more anchors (majority does not guarantee being a maximum).
These two flaws make paper's sole theoretical contribution unsubstantiated.

## Empirical
-   The paper prominently claims a 276x speedup over Minimap2. However, this is based on a somewhat misleading comparison between NAL running on a GPU, and Minimap2 running on a CPU.. Comparing fundamentally different hardware platforms significantly inflates the perceived advantage. The paper's own Figure 5 shows comparisons GPU implementations of Minimap2, such as Parabricks (NVIDIA Parabricks, mm2-gb). Against these comparable baselines, NAL's speed advantage is drastically less. It is in fact slower than Parabricks for sequences up to 5k length, and the speed up gain for the longer sequences seems to be in the [1.5x-2x] range for longer and being slower for shorter than 5k lengths. These are drastically less than the headlined 276x figure. Omitting this crucial context significantly misrepresents and over states the paper's empirical contribution regarding performance.

- Missing baselines: The paper claims SOTA but also fails to compare against published, high-performance GPU implementations of BWA-MEM (e.g., Pham et al., 2023), the other major family of sequence aligners alongside Minimap2. Pham et al. also demonstrated significant speedups for their GPU BWA-MEM over the highly optimized CPU version (BWA-MEM2). Pham et al. even report a higher throughput than Parabricks. See this part in Section 4.2 of the paper: "Notably, we compared our implementation against Clara Parabricks, NVIDIA's proprietary implementation of BWA-MEM on GPUs...". Therefore, to make a credible. Omitting GPU BWA-MEM represents an incomplete benchmark suite and leaves the SOTA claim unsubstantiated

## Minor points
* The paper slightly misrepresents Minimap2's capabilities by suggesting its exact-match seeding simply "breaks down" with mismatches; Minimap2 incorporates heuristics specifically designed to handle noisy reads effectively.
* The paper motivates its simple RC-invariance approach (NAL-RC) by claiming a potential downside ("may break translation continuity") of a competing architectural method (Caduceus), but this claim is not supported by the cited source (Schiff et al., 2024).
* The accuracy comparison uses Minimap2, which has known limitations in repetitive regions addressed by successors like Winnowmap2 (Jain et al., 2020). The evaluation doesn't show if NAL overcomes these specific baseline limitations.

**Questions:**

I am open to hearing authors responses to any of the questions raised above

---

### Official Review · Reviewer_gBR8 · 2025-11-01

**Soundness:** 2
**Presentation:** 1
**Contribution:** 3
**Rating:** 4
**Confidence:** 3

**Summary:**

NeurALigner proposes an embedding-based DNA sequence alignment, leveraged with vector databases and an adaptive seed allocation algorithm. The embeddings are generated with neural networks. It enables an efficient search by indexing reference regions in a sparse manner, and matching reads based on embedding similarity. This makes the matching more robust and fast. NeurALigner shows significant acceleration compared to existing baselines, and shows highest accuracy on DNA sequence alignment on generated long reads.

**Strengths:**

**[S1]** NeurALigner leverages sequence embeddings in a vector space for sequence alignment, and provides an end-to-end solution for embedding-based DNA sequence alignment on GPUs.

**[S2]** The proposed adaptive seed allocation algorithm is unique and works as a solution for difficult alignments.

**[S3]** NeurALigner works well in experiments, and is significantly faster than the baseline methods by sparse indexing of seeds.

**[S4]** The preliminary for the whole seed-chain-extend algorithm was described in detail. This can help readers who are new to the domain.

**Weaknesses:**

**[W1]** Some main contributions seem to be misleading or overstated.
NeurALigner claims to be a GPU-optimized framework, however, there are no specific design components.
The suitability for GPUs rather seems to be an effect of using neural networks and FAISS, not a contribution. This effect is valid for any method that uses such components, not something that NeurALigner newly introduces or optimizes.
Also, it seems that NeurALigner gains speed and scalability not through GPU optimization but through sparse indexing.
The mismatch tolerance obtained by leveraging embedding vectors was already done in prior works.
 To present as an advantage of NeurALigner, it should have a distinctive aspect compared to works like ESA.


**[W2]** Experiments are insufficient to prove the claims in the paper.
The whole experiment seems to be done in simulated data, not a real query read dataset. It seems like NeurALigner may need long reads to obtain reasonable accuracy. Is this a drawback? Some discussion can be added.
If not, experiments with real-world queries can be done to ensure generalization.
Also, sources of reads are encouraged to be in the main paper.
A more detailed analysis between ESA and NeurALigner is encouraged. Also, it is unclear why the length of the read inhibits the comparison of alignment quality.
More comparison with prior works can be done, other than Hyena-DNA, NT and minimap2. BWA and ESA can still be compared in the aspect of mapping and alignment. Also, more comparisons with DNA model architectures are encouraged. Currently, Table 2 is mixed with an ablation study of the encoder.
Do the embeddings actually obtain transfer continuity? Experiments or visualizations about such an aspect are encouraged, as the method was built upon the assumption.


**[W3]** The overall paper has presentation issues that should be improved, as listed below.
The main methodology section should be self-contained.
Readers should understand the encoder architecture without referring to Hyena-DNA. There is too little information of what the encoder architecture is.
The overall explanation is vague.  (e.g., it is unclear how the vector database is constructed, how similarity is computed during approximate nearest neighbor search, how candidate selection is performed during chaining, and what threshold is used for adaptive seed allocation.)
Some of the details in Appendix A.3-A.4 are encouraged to be in the main paper. They contain information of how the actual indexing and chaining is done regarding a vector database.
The presentation of Section 3 lacks organization.
Prior works, the proposed NeurALigner, experimental details, and standard pipelines are all mixed in the sections. It can be arranged into separate sections for clarity.
(e.g., Section 3.1.1: preliminary, prior methods, and the proposed process are all presented together.)
The claimed challenges lack details.
The current explanation is too vague to understand what the challenge is, and how the problem occurs. It should provide such details for readers to figure out how NeurALigner can address the challenges.
They are briefly mentioned in the 3rd passage, and are not illustrated further.
saw some typos:
Sub-title typo in “Encoder” at Section 3.1
Section 5 is likely to be “Conclusion”, not “Discussion”.
is seed-chain-extend,  seed-and-extend, and seed-chain-align different mechanisms? If not, please use just one term.

**Questions:**

- Do the embeddings actually obtain transfer continuity?

- Does NeurALigner only perform well on long reads? Can it generalize to real-world read data?

- Do simulation data differ much to the real-world data? To what extent do the lengths of the generated reads differ from those of real-world reads?

- What is the main difference between NeurALigner and ESA?

- What is NAL-MLM in Figure 2? Why is it being trained with CE Loss yet never mentioned in the main methodology section?

- Why are there no descriptions of “DNA sequence models”? They are primary works that enable sequence embeddings in metric space as encoders, but there is no information other than Hyena-DNA.

There are some suggestions related to the weaknesses in writing.

- Please separate the context of related works, preliminaries, and the proposed method.

- Figure 2 can be leveraged more to show the readers the overview, yet it was hardly used. The current version lacks the big picture explanation of how the overall NeurALigner operates end-to-end, before going in to the details.

---

### Note · Authors · 2025-11-24

I have read and agree with the venue's withdrawal policy on behalf of myself and my co-authors.